# Benchmarking the negatives: Effect of negative data generation on the classification of miRNA-mRNA interactions

**Efrat Cohen-Davidi, Isana Veksler-Lublinsky** [ID] *

Department of Software and Information Systems Engineering, Ben-Gurion University of the Negev, Beer-Sheva, Israel

* vaksler@post.bgu.ac.il

## Abstract

MicroRNAs (miRNAs) are small non-coding RNAs that regulate gene expression post-transcriptionally. In animals, this regulation is achieved via base-pairing with partially complementary sequences on mainly 3' UTR region of messenger RNAs (mRNAs). Computational approaches that predict miRNA target interactions (MTIs) facilitate the process of narrowing down potential targets for experimental validation. The availability of new datasets of high-throughput, direct MTIs has led to the development of machine learning (ML) based methods for MTI prediction. To train an ML algorithm, it is beneficial to provide entries from all class labels (i.e., positive and negative). Currently, no high-throughput assays exist for capturing negative examples. Therefore, current ML approaches must rely on either artificially generated or inferred negative examples deduced from experimentally identified positive miRNA-target datasets. Moreover, the lack of uniform standards for generating such data leads to biased results and hampers comparisons between studies. In this comprehensive study, we collected methods for generating negative data for animal miRNA–target interactions and investigated their impact on the classification of true human MTIs. Our study relies on training ML models on a fixed positive dataset in combination with different negative datasets and evaluating their intra- and cross-dataset performance. As a result, we were able to examine each method independently and evaluate ML models' sensitivity to the methodologies utilized in negative data generation. To achieve a deep understanding of the performance results, we analyzed unique features that distinguish between datasets. In addition, we examined whether one-class classification models that utilize solely positive interactions for training are suitable for the task of MTI classification. We demonstrate the importance of negative data in MTI classification, analyze specific methodological characteristics that differentiate negative datasets, and highlight the challenge of ML models generalizing interaction rules from training to testing sets derived from different approaches. This study provides valuable insights into the computational prediction of MTIs that can be further used to establish standards in the field.

**Data Availability Statement:** Code developed in this study is available on GitHub at https://github.com/IsanaVekslerLublinsky/MTI_negatives (v1.0.0). All datasets generated in this study,

including training and testing sets, are available in Zenodo repository: https://doi.org/10.5281/zenodo.10454080.

**Funding:** This work was supported by the Israel Science Foundation (ISF 520/20) received by IVL. The funders had no role in study design, data collection and analysis, decision to publish, or preparation of the manuscript.

**Competing interests:** The authors declare that they have no competing interests.

## Author summary

Gene expression regulation is fundamental for all organisms' development, homeostasis, and environmental adaptation. microRNAs (miRNAs) play a central role in post-transcriptional gene regulation by binding to target mRNAs and repressing their translation or mediating their degradation. Technical challenges in experimental miRNA target identification led to growing interest in computational target prediction. While machine learning (ML) models have shown success in this area, they rely heavily on artificially generated negative examples due to limited experimental data. The diversity of methods for generating negative interactions and the lack of a uniform standardized approach introduce bias and hinder the comparison of results across different studies.

In this study, we collected methods for generating negative data for animal miRNA–target interactions and analyzed their impact on classifying true interactions in humans. Using an ML approach, we evaluated miRNA–target prediction performance within and across negative datasets. We also analyzed unique features distinguishing between the negative datasets to understand the performance results better. Our study shows that negative data is essential for accurately classifying miRNA–target interactions and that ML models struggle to apply what they have learned from the training set when faced with new data derived from different approaches. This study particularly appeals to researchers interested in miRNA–target classification. It emphasizes the need for standardized methods to enhance comparability between studies.

## 1 Introduction

MicroRNAs (miRNAs) are small non-coding RNAs that exist in the genomes of animals, plants, and some viruses [1], playing key roles in regulating gene expression post-transcriptionally. miRNAs are generated through a multi-stage process by endogenous protein factors [2]. Upon maturation, miRNAs associate with Argonaute proteins to form the miRNA-induced silencing complex (miRISC). Animal miRNAs recognize and bind partially complementary sequences mainly on the 3' untranslated region (3'UTR) of target mRNAs, leading to translational inhibition and/or degradation of the targeted mRNAs [3]. This binding mode allows one miRNA to potentially regulate multiple target mRNAs and for a single mRNA to be targeted by multiple different miRNAs. In contrast, plant miRNAs predominantly have high complementarity to unique sites within the coding region, which promotes target cleavage by miRISC [4].

Animal miRNAs have diverse functions in development and physiology and human miRNAs have been implicated in many diseases. Understanding the molecular mechanisms and the functional roles of miRNAs is essential for unraveling the complex regulatory networks that govern gene expression and for developing novel therapeutic strategies for a variety of diseases [5].

Identifying miRNA target sites on mRNAs is crucial for understanding the involvement of miRNAs in cellular processes. Various high-throughput experimental methods have been developed over the years to identify miRNA-target interactions (MTIs) [6, 7]. Early methods measured changes in mRNA levels after miRNA over-expression or inhibition in tissue-cultured cells [8]; however, they were limited by noise from indirect miRNA regulation and unknown binding site sequences [6, 7]. Later, crosslinking and immunoprecipitation (CLIP) methods, such as HITS-CLIP [9, 10] and PAR-CLIP [11], were developed to capture miRISC

bound miRNAs and mRNA targets, however, they did not provide the binding relationships which need to be computationally predicted. More recently, advanced methods, such as CLASH [12], CLEAR-CLIP [13, 14], and modified iPAR-CLIP [15], have been developed to capture miRNAs bound to their direct targets (recovered as chimeric reads). Due to technical challenges, these methods have so far been applied to a limited number of model organisms, thereby promoting the use of computational predictions to expand miRNA–target repertoires.

Over the years, many tools for MTI prediction in animals have been developed [16], relying on determinants such as base-pairing patterns (primarily in the seed region), thermodynamic pairing stability, target site conservation and accessibility, proximity to 3'UTR ends, nucleotide composition, etc [17]. Recently, machine learning (ML)-based methods have been introduced to distinguish positive miRNA-mRNA pairs (those detected as interacting in experiments) from negative pairs (those with no evidence of interaction), utilizing some of these determinants as features. These approaches exhibit variations in ML techniques, feature selection, dataset choice, and negative data generation (e.g., [18–22]). In our previous study [23], six widely used ML models in computational biology were examined for their performance in the binary classification of MTIs. The XGBoost classifier consistently outperformed other models across all datasets, underscoring its effectiveness in MTI prediction tasks.

Most of the published experimental MTIs represent positive data. At the same time, no high-throughput biological assays are available to capture negative instances, posing a significant challenge to the development of ML models. Consequently, ML approaches are compelled to rely on artificially generated or inferred negative interactions [24]. The inferred negative (non-interacting) miRNA-target pairs are those that have not been identified as positive in experimental positive miRNA-target datasets; however, it is important to note that positive interactions were identified under specific conditions (i.e., cell type, developmental stage), which may not fully represent the broader context of miRNA-target interactions. The selection of an appropriate method for generating negative examples plays a crucial role in the predictive ability of the model. Striking the right balance is imperative, as excessively distinctive or highly similar negative examples compared to positive ones may hinder the efficient training of the model in discriminating between positive and negative instances (potentially leading to overfitting or underfitting, respectively) [25].

So far, different studies have used various approaches to generate negative interactions, with no standardized method, limiting the ability to compare results across studies. Differences are observed in several aspects, including the data resources for negative interactions, the use of original or artificial sequences, and the criteria for selection among multiple candidates. Briefly, mirMark [20] and DeepMirTar [26] used CLASH data and generated negative examples by shuffling the original miRNA in each positive interaction. mirTDL [27] and miRAW [28] utilized inferred interactions from experimental data. chimiRic [18] employed CLIP data to generate interactions between recovered miRNAs and mRNAs, that were not observed in positive datasets (e.g., CLASH). TarPmiR [29] and MirTarget [21] generated negative examples from CLASH interactions by searching for alternative sites within the full 3' UTR region of the original target; however, their approaches to creating interaction pairs and filtering less favorable interactions varied.

An alternative approach to address the shortage of negative data is to utilize one-class classification (OCC) models [30] which operate by using single-class examples. OCC has proven effective in solving real-life problems characterized by abundant data for one class and limited data for other classes. This scenario frequently arises in anomaly detection tasks, where the objective is to identify outliers [31]. In such models, only samples from one class are utilized during the training process, such that the ML model learns to distinguish them from other classes in the test set. Two models that belong to OCC are one-class SVM, an algorithm that

learns a decision function for novelty detection [32], and Isolation Forest, a tree-based anomaly detection method [33]. In a study by Cárdenas et al. [34], OCC was used to predict the binding of host human miRNAs to the SARS-CoV-2 RNA sequence. The one-class SVM outperformed multi-class models (SVM and Random Forest) in predicting the binding of miRNAs to immune genes, considering the SARS-CoV-2 5'-UTR region. These OCC models offer potential solutions for MTI prediction tasks, which have to be evaluated for their effectiveness in learning interaction rules without relying on explicit negative interaction examples.

In this comprehensive study, we investigated how different approaches to generating negative data impact the classification of true human MTIs. We utilized datasets of direct high-throughput MTIs as positive examples and implemented various methods to generate datasets of negative interactions. For each dataset, we trained and tested ML classifiers to predict miRNA–target interactions within the same dataset. To assess the sensitivity of the trained models to the negative data, we also evaluated cross-dataset classification performance. Additionally, we conducted a feature importance analysis to elucidate the differences between the methods and to uncover the specific factors that contribute to the distinction between classes. Moreover, we explored two OCC models that utilize solely positive interactions for training. The results of this study provide valuable insights into the computational prediction of MTIs, which can be further used to establish standards in the field.

## 2 Materials and methods

In this work, we evaluated binary classifiers trained on a fixed human positive dataset of MTIs and negative datasets generated by different methods, where each interaction is characterized by a set of features. In addition, we examined two one-class classification models that learn to classify examples by using only positive interactions for training. Importantly, positive MTIs are those that were recovered in high-throughput experiments such as CLASH (see below). Negative MTIs are those that have no experimental evidence of binding.

### 2.1 Software packages and tools

The code developed during this research was implemented in Python, running on a Linux platform and employing bioinformatics, data analysis, and ML packages. Specifically, the bioinformatics packages include ViennaRNA (v2.5.0) [35], Biopython (v1.80) [36], bowite2 (V2.3.5.1) [37], cutadapt (v3.5) [38] and NCBI Blast [39]; the data analysis packages include pandas (v1.3.4) [40] and numpy (v1.21.5) [41]; ML packages include scikit-learn (v1.0.2) [42], XGBoost (v1.7.1) [43], and the SHAP values packages (v0.41.0) [44].

### 2.2 Definitions

For each MTI, we define the following terms regarding the target molecule: **fragment** is the target region that is obtained in experiment (e.g., CLASH, see below) and is used to calculate the interaction; **site** is the actual binding region within the fragment obtained by duplex calculation; **full mRNA** refers to the 3'UTR sequence that contains the target site; interaction **duplex** represents the binding pattern between a miRNA and a site sequence and is calculated by RNAduplex tool from ViennaRNA suite [27]. We classified the duplexes based on their seed type: *canonical seed*, *non-canonical seed*, or *other*. Canonical seed interactions are defined as exact Watson-Crick pairing in positions 2–7 or 3–8 of the miRNA. Non-canonical seed interactions may contain GU base pairs and up to one bulged or mismatched nucleotide at these positions [12]. Given the existing focus of works in this field on canonical and non-canonical interactions, we opted to specifically explore these two seed types of interactions, collectively referred to as **valid** in our study. Our objective was to analyze high-quality

interactions, ensuring that negative interactions are comparable to positive interactions and not overly artificial. Consequently, for negative interactions, we considered only valid interactions, as well.

## 2.3 Data retrieval and processing

Precursor and mature human miRNA sequences were downloaded from miRBase (releases 17–22) [45]. Human 3'UTR sequences were downloaded from the Ensembl Biomart database [46], as in our study, only interactions that fall in 3'UTR are considered.

### CLASH

CLASH-based techniques, that are derived from AGO-CLIP methods, use an extra step to covalently ligate the 3' end of a miRNA and the 5' end of the associated target RNA within the miRISC. Subsequent cloning and sequencing of isolated chimeric miRNA–target reads facilitate the identification of direct MTIs. The portion of chimeric reads in such sequencing data is about 2%. The rest of the reads represent separate miRNAs and mRNA target fragments, as in CLIP studies. In CLIP datasets, determining the specific miRNA involved in each interaction relies on bioinformatic predictions [47, 48].

We denote by *CLASH*, the high-throughput datasets composed of chimeric MTIs retrieved by methods e.g., CLASH, CLEAR-CLIP. In our study, we used four such datasets generated from human cells (*h1-h4*, Table 1); where each dataset contains miRNA–fragment pairs. All datasets were processed as described in [23], considering only interactions that involved the 3'UTR region. The processing involved converting the datasets into a consistent format where each record includes: miRNA name, miRNA sequence, target fragment sequence, full mRNA sequence, and the fragment's coordinates within the mRNA.

### CLIP

We downloaded the complementary CLIP data for the dataset *h3*, generated by CLEAR-CLIP experiment in Huh-7.5 cells [13] (NCBI GEO database series GSE73059). This data provides separate miRNA and mRNA sequences associated with AGO complexes, such that the interacting miRNA-mRNA pairs are unknown. We developed a pipeline to match the sequences to the respective names of miRNAs and mRNAs. First, we removed the adapter sequences (as in [13]) and filtered out reads shorter than 18nts. Next, we created two index databases for 3'UTR sequences and miRNA precursor sequences using bowtie2 [37], and aligned the reads against the databases (resulting in miRNA.SAM and mRNA.SAM files, respectively). From the miRNA.SAM file, we further removed sequences longer than 25nts [50]. To obtain the mature miRNA name corresponding to the aligned region, we mapped the available mature miRNA sequences from miRBase against the precursors as well and selected the mature miRNA with the highest overlap.

From the mRNA.SAM file, we removed sequences shorter than 40 or those that mapped with mismatches. In cases where a read was aligned to multiple 3'UTR sequences, we kept the

**Table 1. High-throughput datasets of human chimeric MTIs.**

| Dataset | Cell type/developmental stage | Experimental method | Reference |
|---|---|---|---|
| **h1** | Embryonic kidney 293 cells | CLASH | [12] |
| **h2** | A mix of 6 datasets | AGO-CLIP endogenous ligation | [15] |
| **h3** | Hepatoma cells (Huh-7.5) | CLEAR-CLIP | [13] |
| **h4** | Melanoma cells | qClash | [49] |

match to the longest one. Based on the alignment, we extended each fragment to 75 nts (similar to [13]). For each 3'UTR, we kept one alignment among those that overlapped by more than 85%. These steps resulted in two lists *miRNA-clip* and *mRNA-clip*, composed of miRNAs (name and sequence) and mRNAs (name, fragment sequence, and full mRNA), respectively.

### TarBase

We downloaded experimentally labeled negative interactions from *TarBase* v8 [24]. Each interaction record in TarBase dataset consists of gene ID, miRNA name, source species, tissue, cell line, and experimental method. We discarded all non-human or unknown cell line records. We filtered duplicate records of the same miRNA-gene pair. Then, we supplemented each record with the corresponding miRNA and 3'UTR sequence from miRBase and Ensembl, respectively, resulting in a dataset denoted as *TarBase*, which contains miRNA-full mRNA pairs.

### 2.4 Generating interactions

Below, we describe how we used the above resources to generate positive and negative interactions for our analysis.

### Positive interactions

Datasets *h1-h4* are considered positive, as they contain experimentally identified miRNA-mRNA pairs. For all records in these datasets, interaction duplexes were calculated and classified and only valid interactions were kept (Table 2). In addition, we filtered duplicate interactions (i.e., the same miRNA, target gene, and duplex). We chose *h3* as the positive interactions dataset for our experiments. In addition, we created *full-positive-dataset* (FPD) from *h1-h4*, which was used to filter negative interactions produced by various methods, as described below.

### Negative interactions

We collected methods for generating negative interactions from the literature (refer to Fig A in S1 Text for details on the comprehensive literature search). We grouped methods that share a common idea and utilize similar sources of information, although the implementation details may vary across studies. Below, we provide relevant details for each method and summarize all methods in Fig 1. Furthermore, the pseudocode of the algorithms is provided in S1 Text.

**Method 1: Mock miRNA.** A mock miRNA is a random shuffling of a real mature miRNA sequence; for each positive interaction in *h3*, we employed a random shuffling (1-mer/mono, using uShuffle with k = 1 [51]) to the original miRNA sequence until the sequence at positions 2–7 and 3–8 in the shuffled miRNA did not match with the same regions of any real human miRNA (similar to, e.g., [20, 23]). Next, we provided the mock miRNA and the full mRNA sequence as inputs to RNAduplex and repeated the process until a valid duplex was obtained

**Table 2. Summary of positive datasets processing pipeline.**

| Dataset | No. of interactions (original publication) | No. of interactions in 3'UTRs | Final dataset (valid interactions) |
|---|---|---|---|
| h1 | 18,514 | 8,796 | 4,991 |
| h2 | 10,567 | 5,844 | 4,170 |
| h3 | 32,712 | 5,597 | 3,517 |
| h4 | 237,767 | 114,188 | 77,326 |

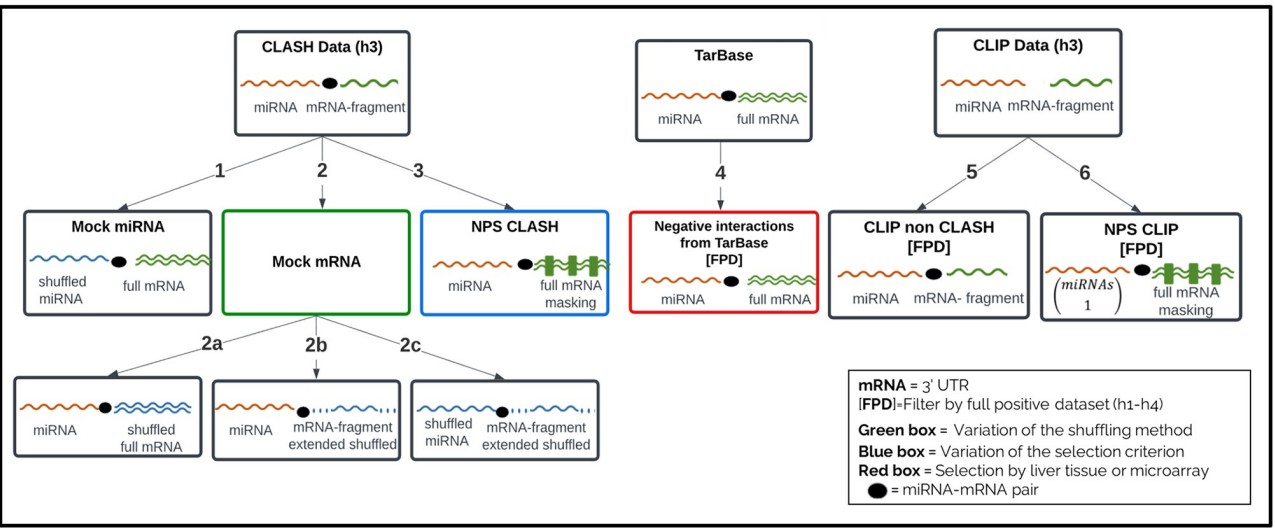

**Fig 1. Summary of methods for generating negative interactions utilized in this study.** The first three methods use the positive dataset *h3* (miRNA-mRNA fragment-full mRNA records) to produce a corresponding negative interaction for each positive interaction. The fourth method uses TarBase data (miRNA-full mRNA records), and the last two methods employ complementary CLIP data of *h3* (*miRNA-clip* list, *mRNA-clip* list). The TarBase and CLIP-based methods utilize the full-positive dataset (FPD) to filter genuine interactions from potential negative interactions. Note that the black ellipse in the figure represents miRNA–target pairs; but it does not appear for the CLIP dataset, which provides separate miRNA and mRNA sequences.

or the number of attempts reached a predefined limit. Successful attempts involve generating artificial miRNA sequences, which, although they do not naturally occur in human cells, form valid interaction duplexes. We anticipate detecting distinct feature patterns between genuine and artificial miRNA interactions. The dataset resulting from this process is referred to as *Mock_miRNA* (Table 3).

**Method 2: Mock mRNA.** A mock mRNA is generated by randomly shuffling a specific region of the full mRNA sequence. For each positive interaction in *h3*, we investigated three

**Table 3. Summary of the negative datasets generated by different methods.** The method numbers correspond to method numbers in Fig 1. FPD denotes the *full-positive-dataset*.

| Method number | Dataset name | Dataset source | Final dataset (valid interactions) |
|---|---|---|---|
| 1 | Mock_miRNA | CLASH (h3) | 3,517 |
| 2a | Mock_di_mRNA | CLASH (h3) | 3,517 |
| 2a | Mock_mono_mRNA | CLASH (h3) | 3,517 |
| 2b | Mock_di_fragment | CLASH (h3) | 3,517 |
| 2b | Mock_mono_fragment | CLASH (h3) | 5,517 |
| 2c | Mock_di_fragment_&_miRNA | CLASH (h3) | 3,517 |
| 2c | Mock_mono_fragment_&_miRNA | CLASH (h3) | 3,517 |
| 3 | NPS_CLASH_MFE | CLASH (h3) | 3,379 |
| 3 | NPS_CLASH_Random | CLASH (h3) | 3,379 |
| 4 | TarBase | TarBase + FPD | 31,526 |
| 4 | TarBase_Liver | TarBase + FPD | 3,706 |
| 4 | TarBase_microarray | TarBase + FPD | 24,871 |
| 5 | CLIP_non_CLASH | CLIP (h3) + FPD | 194,010 |
| 6 | NPS_CLIP_MFE | CLIP (h3) + FPD | 3,517 |
| 6 | NPS_CLIP_Random | CLIP (h3) + FPD | 3,517 |

sub-cases: (2a) randomly shuffling the full mRNA sequence; (2b) extending the target site sequence 100 nts on each side, shuffling the extended fragment, and then trimming 50 nucleotides on each side (as these will be used for context features, see below); (2c) randomly shuffling both the mRNA sequence as in (2b) and the miRNA as described above. For all strategies, we provided the mock mRNA and the miRNA sequences as inputs to RNAduplex. We repeated the process until a valid duplex was obtained or the number of attempts reached a predefined limit. Similarly to the mock miRNA strategy, through the simulation of artificial mRNA sequences that produce valid interactions, we anticipate observing distinct features in interactions involving bonafide versus artificial mRNA sequences. We employed two types of shuffling of the mRNA: nucleotide (1-mer/mono) and dinucleotide (2-mer/di) shuffling, using the uShuffle tool, with k = 1 and k = 2, respectively. K-mer shuffling involves randomizing the order of nucleotide sequences that preserve the exact k-mer counts. The k = 2 option was examined to accommodate the mRNA composition features used in this study (see Features). In total, we generated six datasets (three sub-cases with two types of shuffling), which are denoted as *Mock_mono_mrna*, *Mock_di_mrna*, *Mock_mono_fragment*, *Mock_di_fragment*, *Mock_mono_fragment_&_miRNA* and *Mock_di_fragment_&_miRNA* (Table 3).

**Method 3: Non-positive sites CLASH (NPS CLASH).** Given an interacting miRNA–mRNA pair, this method identifies alternative binding regions of the miRNA on the full mRNA that do not overlap with the positive sites of this pair (similar to, e.g., [19]). To that end, we first grouped the CLASH records of dataset *h3* according to miRNA and mRNA names. Then, for each miRNA–mRNA pair, we masked on the full mRNA all positive sites observed for this pair (extended by 10nt per side) with "N" to avoid interactions in these regions. The masked full mRNA was scanned with a sliding window of 75nt (similarly to Methods: 2.3: CLIP) with steps of 40nt. For each window, we calculated the interaction duplex with the miRNA and kept only valid candidates. In instances where multiple candidates were found for the pair, we used one of the two selection options: choosing the candidate with the lowest minimum free energy (MFE) or opting for random selection. Collectively, this method identifies high-quality alternative binding regions that could potentially accommodate the miRNA, even though the actual binding recovered by the CLASH experiment occurred in a different region. We refer to these datasets as *NPS_CLASH_MFE* and *NPS_CLASH_Random*, respectively (Table 3).

**Method 4: Negative interactions from TarBase.** In this method, we used *TarBase* dataset, consisting of experimental negative data interactions in the form of miRNA-full mRNA pairs (similar to, e.g., [22]). We filtered pairs that are present in *FPD* and generated the following three sub-datasets: (i) the full dataset; (ii) to examine whether the tissue of origin impacts the results, we kept only interactions that originate from the liver tissue, as this is the tissue of origin for the positive interactions dataset *h3*; (iii) we kept only interactions derived from miRNA over-expression microarray data. In such experiments, mRNAs whose expression does not change when miRNAs are over-expressed are considered negative targets. In each sub-case, we calculated the interaction duplex using miRNA and full mRNA sequences and kept only valid interactions. We refer to the generated datasets as *TarBase*, *TarBase_Liver*, *TarBase_microarray*, respectively (Table 3).

**Method 5: CLIP non CLASH.** The objective of this method is to generate valid interactions between miRNAs and mRNA fragments that were observed separately in the CLIP data such that interactions between the miRNA (or its seed family members) and the full mRNA were not recovered in any available CLASH experiment (similar to, e.g., [18]). Given two lists (*miRNA-clip*) and (*mRNA-clip*), we first generated all possible miRNA-mRNA pairs from the lists. We then removed pairs for which an interaction of the same mRNA and a miRNA from the same seed family exists in the *FPD*. We calculated the duplex between the miRNA and the

mRNA fragment and kept only valid interactions. We noticed that forcing duplex generation between non-necessarily interacting miRNA and mRNA fragments resulted in some valid interactions (canonical/non-canonical seeds) with a long unpaired tail at the 3'end of a miRNA. To eliminate this bias, we discarded interactions with a tail larger than 5nt. We denote the dataset of remaining interactions as *CLIP_non_CLASH* (Table 3).

**Method 6: Non-positive sites CLIP (NPS CLIP).**   This method generates new interactions for mRNAs found in CLIP data; such that the new binding sites do not overlap with known interaction sites (similar to, e.g., [21]). We extracted relevant full mRNAs from *mRNA-clip* list of *h3* CLIP dataset. Then, we masked regions that corresponded to fragments from *h3* CLIP dataset and the *FPD* (extended by 10nt per side) and applied a sliding window of 75nt with a step of 40nt to generate candidate negative fragments (similarly to Method 3). For each window, we randomly selected a miRNA from *miRNA-clip* and calculated the interaction duplex. We repeated the process until a valid duplex was obtained or the number of attempts reached a predefined limit. In case multiple candidates were found for an mRNA, we employed one of two selection methods: interaction with the lowest MFE or random selection. This method identifies high-quality alternative binding regions on targeted mRNAs that could potentially accommodate miRNAs expressed in the same cell type, even though these regions were not identified in the examined CLIP or available CLASH datasets. We refer to these datasets as *NPS_CLIP_MFE* and *NPS_CLIP_Random*, respectively (Table 3).

## 2.5 Features

To represent positive and negative MTIs, we used 500 features, of which 490 features were adopted from our previous study [23] and the remaining 10 features that characterize miRNA pairing were added in this study. The classification of the features into five categories is shown in Table 4. For the full list of features and their description, see S1 Text and S1 Table.

## 2.6 Machine Learning methods

We employed the following ML techniques in our experiments.

**2.6.1 Binary classification with XGBoost.**   To classify MTIs into positive or negative, we utilized eXtreme Gradient Boosting (XGBoost) [43], based on its superior performance in a similar task demonstrated in our previous study [23]. XGBoost is a feature-based ensemble learning algorithm rooted in the principles of gradient boosting, seamlessly combining weak learners to construct a highly accurate and robust predictive model. It is recognized for its efficiency, speed, and versatility in excelling at classification and regression tasks. Additionally, XGBoost offers interpretability features, providing a nuanced understanding of model decisions. We performed hyperparameters optimization to the XGBoost model by using Grid Search CV with five-fold cross-validation.

**Table 4. Categories of features that are used to represent MTIs.**

| Category | Number of features | Description |
|---|---|---|
| Seed features | 13 | Seed composition and properties |
| Free Energy | 7 | Free energy of the duplex and the mRNA at different regions |
| mRNA Composition | 62 | mRNA composition in the site and flanking regions |
| miRNA Pairing | 48 | Binding information at each miRNA position and across the miRNA-target duplex |
| Site accessibility | 370 | Unpaired probabilities of each base at the site region |

**2.6.2 One-class classification models.** One-class classification (OCC) algorithms are widely used for anomaly detection, where data are abundant for one class and limited for other classes. In our study, we tested an OCC-based approach for classifying MTIs by considering negative interactions as outlier observations. To implement this approach, we employed two different methods: One-class SVM [32] and Isolation Forest [33]. One-class SVM is an algorithm that learns a decision function for novelty detection, while Isolation Forest is a tree-based anomaly detection method. We trained these models on positive interactions and evaluated their performance on both positive and negative miRNA–mRNA pairs to identify interactions that deviate from the distribution of positive interactions. In addition, we compared these one-class models with their corresponding binary learning models SVM and Random Forest, respectively. By comparing each one-class model with its binary-class counterpart, we aimed to neutralize potential biases stemming from the choice of algorithm, providing a more accurate evaluation of the models' effectiveness in one-class classification scenarios.

## 2.7 Data leakage, imbalance and splitting into training and testing sets

Our ML approach is based on interaction features rather than the interacting sequences themselves. Similar sequences might form different interactions with variations in the duplex structures and their characteristics, and vice versa; different sequences can form duplexes with the same characteristics. Therefore, for each interaction in the negative datasets, we ensured that the positive dataset *h3*, contains no interactions with the same feature values.

Some of the methods generated less or more negative interactions compared to the positive dataset (Table 3). To address the class imbalance problem, we employed the commonly used methods of undersampling, which involves randomly selecting a subset of samples from the larger class.

Accurate determination of the training and testing sets is essential to ensure the reliability of results. Specifically, we ensured that all models were trained and tested on the same positive interactions while varying the negative interactions. We split the positive and negative datasets into 80% training and 20% testing according to a random state (Fig 2). Furthermore, to simulate cross-validation, we repeated the splitting procedure 20 times with different random states, resulting in 20 unique training and testing sets for each dataset. By adopting this comprehensive methodology, we obtained robust and reliable results for our analysis.

## 2.8 Evaluation

We conducted a comprehensive evaluation of the performance of XGBoost in classifying MTIs intra- and cross-negative datasets (Fig 2). Training and testing sets derive from the same negative dataset in intra-dataset classification or two different datasets in cross-dataset classification. Thus, we enumerated all possible pairs of training and testing datasets: ($train_i$, $test_j$) where $1 \leq i,j \leq N$ and $N$ is the number of methods to generate negative interactions. For each pair, we trained 20 XGBoost classifiers with the training sets of dataset $i$ (corresponding to 20 splits) and evaluated their performance on the testing sets of dataset $j$. We calculated the mean and the standard deviation values of the classification accuracy (ACC), True Negative Rate (TNR), True Positive Rate (TPR), and F1 score.

Additionally, to ensure a fair comparison of one-class SVM with SVM model and of Isolation Forest with Random Forest model, we executed the GridSearch algorithm with five-fold cross-validation to determine the optimal hyperparameters for the one-class models, while ensuring compatibility with the hyperparameters of the binary-class models. For the training of the binary-class models, we used the training sets generated above, while for the one-class models, we used partial training sets that included only the positive interactions. All the

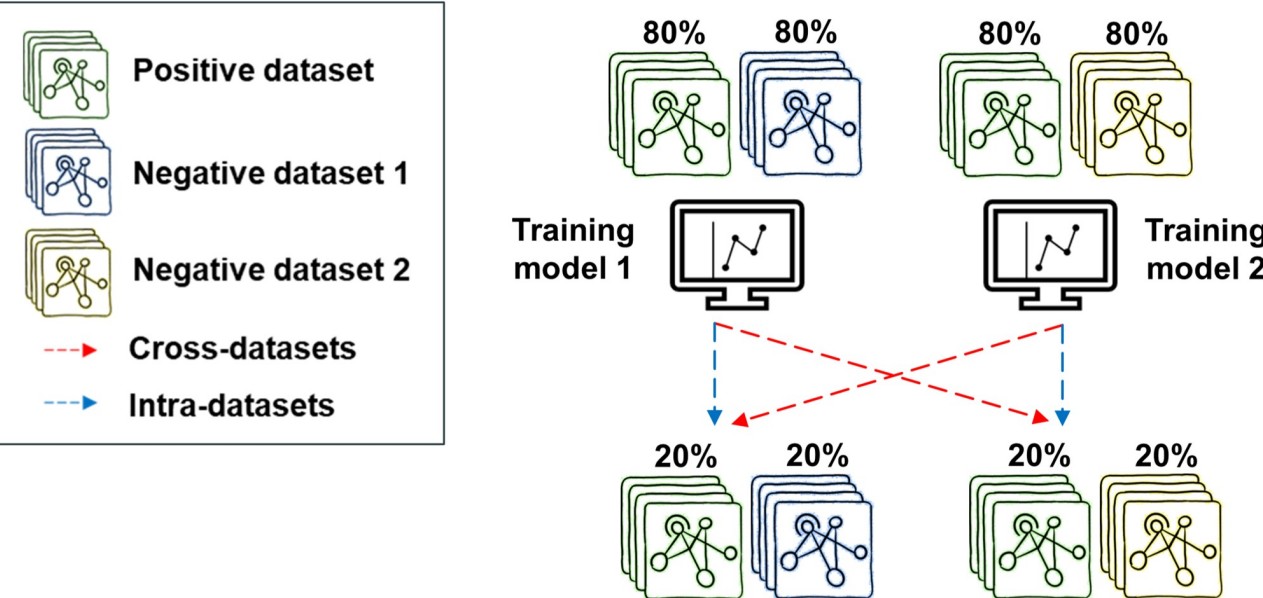

**Fig 2. Evaluation of negative data generation methods by intra- and cross-datasets analysis.** The models were trained and tested on identical positive interactions, while the negative interactions varied. The positive and negative datasets were randomly split into 80% for training and 20% for testing. Training and testing sets derive from the same negative dataset or two different datasets in intra- or cross-dataset classification analysis, respectively. To simulate cross-validation, we repeated the splitting procedure 20 times with different random states, resulting in 20 unique training and testing sets for each dataset.

models were tested on the same testing sets. Here we computed the mean and the standard deviation of different measures of the 20 tests as well.

## 2.9 Data analysis techniques for interpreting classification model results

We used various techniques to gain a deeper understanding and biological insight into the classification results.

**2.9.1 Identification of the top important features.**   To gain a more comprehensive understanding of the input features and their impact on the classification outcome, we extracted the top important features using the SHapley Additive exPlanations (SHAP) framework. The SHAP approach quantifies the influence of each input feature on the model's classification.

**2.9.2 Calculation of miRNA distribution.**   To analyze miRNA sequence occurrences within a dataset, we generated a cumulative distribution function (CDF). We used the *argmax* function to find the 90% value, which returns the first point in the CDF higher than 90%.

**2.9.3 Calculation of the Kullback-Leibler (KL) divergence.**   The KL divergence is an asymmetric measure calculated on two probability distribution functions of target and source datasets, and measures the distance between them, according to Eq 1.

$$D_{KL}(P||Q) = \sum_{x \in \chi} P(x) log\left(\frac{P(x)}{Q(x)}\right) \tag{1}$$

The KL divergence was employed to assess the information loss between pairs of datasets. $P(x)$ and $Q(x)$ are the miRNA sequence distributions (see Methods: 2.9.2). Notably, $Q(x)$ corresponds to the approximation distribution calculated on the source dataset, while $P(x)$ represents the true distribution derived from the target dataset. $\chi$ denotes the union of miRNA sequences present in both datasets.

**2.9.4 Two negative-classes classification.** We evaluated the ability of a classifier to differentiate interactions originating from two distinct negative datasets as follows. For each pair of negative datasets *(A,B)*, we trained and tested (with 80% and 20% split) an XGBoost model with default hyperparameters such that interactions from A and B were labeled as "1" and "0", respectively.

## 3 Results

In this work, we performed a comprehensive benchmark analysis to examine different methods for generating negative datasets of MTIs. Our methodology relies on training ML models on a fixed positive dataset in combination with different negative datasets and evaluating their intra- and cross-dataset performance. This allowed us to examine each method independently and evaluate ML models' sensitivity to the methodologies utilized in negative data generation (Fig 2). To achieve a deep understanding of the performance results, our evaluation is accompanied by an analysis of unique features that distinguish between datasets.

### 3.1 Intra-dataset analysis identifies inherent biases in some strategies for negative data generation

In this subsection, we evaluate the performance of ML-based binary classifiers to correctly classify positive and negative MTIs derived from the same dataset. For each dataset, we generated 20 random training-testing splits of the data. We then trained XGBoost classifiers on the 20 training sets and measured their average performance in the classification of their respective testing sets (Fig 3). Additionally, we examined sub-cases for some data generation methods to determine their optimal implementation and estimate the contribution of features to

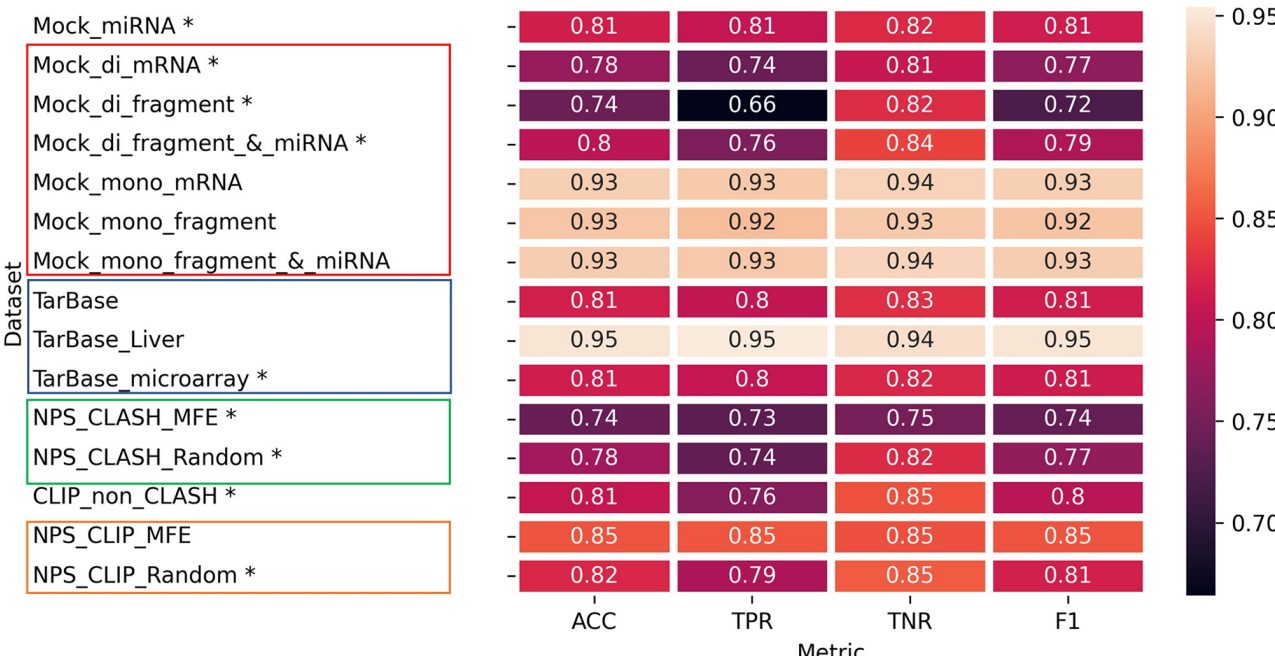

**Fig 3. Evaluation of negative data generation methods based on intra-dataset ML.** The results are organized in a heatmap, where rows correspond to different methods of negative data generation and columns correspond to different performance metrics. Each cell provides the average value obtained from 20 classifiers. The datasets marked with asterisks were chosen for the next step of cross-dataset ML analysis. The boxed methods are sub-cases of the same method. For interpretation of the results refer to feature importance analysis in Fig 4, and Figs B,C in S1 Text.

classifiers' decisions. We found that the performance metrics of **most** datasets were similar and relatively high, ranging as follows: ACC [0.74–0.85], TPR [0.66–0.85], TNR [0.75–0.85], and F1 [0.72–0.85]; however, for some sub-cases, the performance reached much higher values and was subjected to a deeper investigation as described below. Since all metrics (ACC, TPR, TNR, F1) correlated, the primary measure reported below is ACC.

**Mock miRNA.** In this method, interactions are generated for randomly shuffled real mature miRNA sequences. The performance for this method reached ACC of 0.81 (Fig 3), and it is comparable to a previous study [23] that used the same approach of generating negative data and was trained/tested on the same dataset.

**Mock mRNA.** In this method, interactions are generated for randomly shuffled bonafide mRNA target sequences, with variation in the shuffled region (a fragment containing the site or the full mRNA) and the shufflings configuration (1-mer/mono, 2-mer/di). Furthermore, we investigated the integration of mock mRNA with mock miRNA methods. The results show a clear pattern in which *mono* shuffling leads to relatively high performance (ACC of 0.93) while *di* shuffling yields much lower results (ACC of 0.74–0.8) (Fig 3). To gain deeper insights into the contrast between these two shuffling methods, we extracted feature importance lists for the respective classifiers (Fig B in S1 Text). The results indicate that when employing 1-mer shuffling (Fig B(a-c)) in S1 Text, the majority of the top ten essential features are associated with the 2-mer mRNA composition (within the target site or its flanking sequences). This implies that 1-mer shuffling, although it maintains the overall nucleotide frequency, does not maintain the original distribution of 2-mer sequence composition in mRNA sequences. Thus the mRNA composition features play a crucial role in differentiating between positive (original) and negative (shuffled) interactions when using this shuffling type. The sharp reduction of mRNA composition features among the top ten crucial features in 2-mer shuffling (Fig B(d-f) in S1 Text) further supports this observation. Therefore, for cross-dataset analysis, we selected datasets generated with dinucleotide shuffling to avoid biases associated with specific mRNA features.

**Non-positive sites CLASH (NPS CLASH).** The method identifies alternative target sites of a miRNA within the full sequence of its positive target gene. As multiple candidates can be optional for the same miRNA-mRNA pair, we evaluated two selection rules for choosing a site: site with the lowest MFE or random selection, which yielded an ACC of 0.74 and 0.78, for the respective datasets. To further investigate how selection rules may influence cross-classification performance we proceeded with both approaches to cross-dataset analysis.

**Negative interactions from TarBase.** We obtained negative interactions from experimental data available on TarBase. We evaluated the use of the full dataset as well as two filtered datasets (Fig 3). The performance of the classifiers was similar and relatively high for both *TarBase* and *TarBase_microarray* with ACC of 0.81. The highest ACC achieved by the *TarBase_Liver* dataset (0.95) indicates that the model effectively learned specific features that significantly impacted its ability to discern positive from negative interactions (Fig C in S1 Text). The *TarBase_Liver* classifier learned many features that are associated with binding at specific miRNA positions, in contrast to the *TarBase* and *TarBase_microarray* classifiers that had a similar list of top important features that characterize the duplex itself.

To understand this further, we counted the occurrence of each miRNA sequence within the datasets and generated a cumulative distribution function (CDF) (see Methods: 2.9.2). Our analysis revealed that 5%, 20%, 49%, and 33% of miRNA sequences within *TarBase_Liver*, *TarBase*, *TarBase_microarray*, and *h3* are responsible for 90% of the interactions (Fig D in S1 Text). This suggests that the miRNA population in *TarBase_Liver* is not diverse enough, indicating that the classifier learned limited characteristics of negative interactions as opposed to a

positive dataset that is much richer. Therefore, to avoid biases associated with specific miRNA characteristics, we continued with *TarBase_microarray* dataset to cross-dataset analysis.

**CLIP non CLASH.**   The objective of this method is to generate valid interactions between miRNAs and mRNA fragments that were observed separately in the CLIP experiment, such that, interactions between the miRNA (or its seed family members) and the corresponding mRNA were not recovered in the CLASH experiments. This method achieved ACC of 0.81 (Fig 3).

**Non-positive sites CLIP (NPS CLIP).**   This method generates valid interactions between miRNAs and mRNAs found in CLIP data, such that mRNA regions observed in CLIP or available CLASH datasets were masked for binding. As multiple candidates can be optional for any mRNA, we evaluated two selection rules for choosing a site: site with the lowest MFE or random selection, which yielded ACC of 0.85 and 0.82, respectively (Fig 3). Since a similar performance was achieved for both datasets, we chose the one created with a random selection criterion for the cross-dataset analysis.

Overall, intra-dataset analysis helped to identify high-quality datasets that do not suffer from biases introduced by specific strategies (marked with an asterisk in Fig 3), and these datasets are further tested for generalizability to other negative datasets in the cross-dataset analysis below.

**3.1.1 Comparative analysis of key features: Highlighting variations in the classification decision-making process.**   To investigate the difference in the decision-making process of the classification models trained on different datasets of negative interactions, we conducted the following test. For each pair of models, we extracted the union of each dataset's top 100 important features and applied Wilcoxon Signed Rank test on their normalized SHAP values (Fig 4). The decision-making process of the classification models exhibits divergences; notably, methods based on CLIP data (e.g.,*CLIP_non_CLASH*, *NPS_CLIP_Random*) are significantly different from other datasets. Furthermore, we observe notable differences among pairs of sub-cases of the same method (e.g., *Mock mRNA*, *NPS_CLASH*). For example, the significant difference between *NPS_CLASH* methods indicates that selection criterion (MFE or random) among multiple sites affects the impact of different features on the classification outcome. In the random case, miRNA pairing and seed features are the top important features, while in the MFE case, mRNA composition features take precedence.

## 3.2 The cross-dataset analysis demonstrates the models' sensitivity to negative data in the MTI classification task

In the previous subsection, we conducted training and evaluation of dedicated classifiers for each dataset. To further understand how sensitive the models are to different types of negative data, we investigated cross-dataset relationships which are reported in this subsection. To that end, we first characterized the datasets in terms of base-pairing patterns and miRNA distributions. We also compared pairs of datasets by their miRNA distribution using KL distance. Subsequently, we used classifiers trained on specific negative datasets and assessed their classification performance on negative interactions from different datasets (Fig 2). We hypothesized that classifiers that are trained and tested on datasets with similar characteristics would perform better compared to datasets that are not.

**3.2.1 Dataset characteristics.**   First, we classified the interactions (i.e., miRNA-target duplexes) into six classes based on two parameters: (i) seed type, i.e., canonical or non-canonical, and (ii) base-pairing density, i.e., number of base-pairs within the duplex: low <11 bp, medium: 11–16 bp, or high: >16 bp (Fig E in S1 Text). In contrast to the positive dataset, which contains a similar amount of canonical and non-canonical interactions, the negative

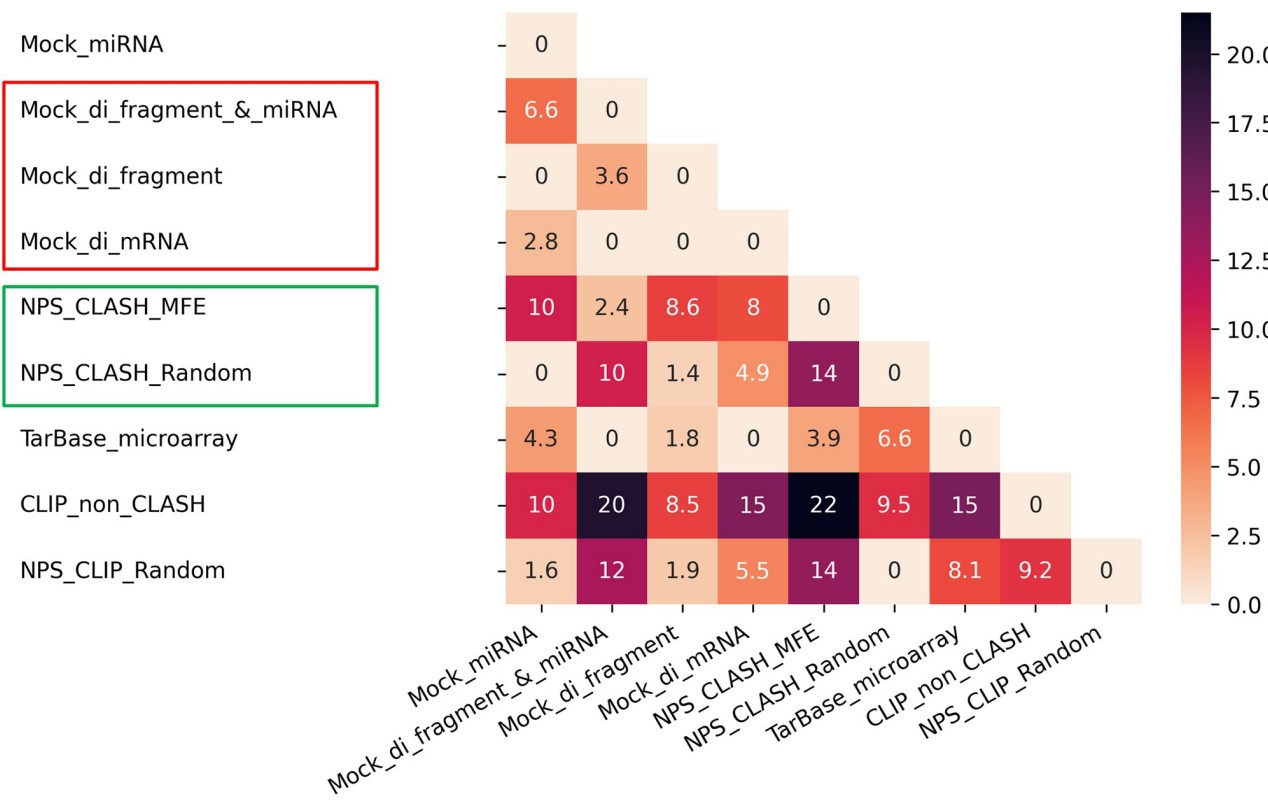

**Fig 4. Comparing top important features of different negative datasets.** Each cell (i,j) represents the -log10(p-value) of the Wilcoxon Signed Rank test applied on the normalized SHAP values of the union of the top 100 features of datasets *i* and *j*. Bonferroni correction was applied to the p-values. Non-significant P-values (greater than 0.05) are marked with 0. The boxed methods are sub-cases of the same method. Included are only methods that were chosen for cross-dataset analysis.

datasets mainly comprise non-canonical interactions. Among all the negative datasets, for both seed types, low base-pairing density interactions represent only a small fraction, and most interactions have medium or high base-pairing density. Furthermore, interactions that are based on clip data (e.g., *CLIP_non_CLASH*, *NPS_CLIP_Random*) exhibit the highest proportion of non-canonical low-density interactions (roughly 10–11%). Furthermore, variations are observed in the proportion of non-canonical interactions with a high or medium density among different datasets. Notably, the *Mock_di_mRNA*, *Mock_miRNA*, *NPS_CLASH_MFE* and *TarBase_microarray* datasets exhibit the highest proportion of high-density interactions (38%-47.1%).

Second, we computed the distribution of miRNA sequences within the datasets by counting the occurrence of each miRNA sequence and generating the cumulative distribution function (CDF) (see Methods: 2.9.2) (Fig F in S1 Text). Our analysis revealed that miRNA appearances in the datasets are non-uniformly distributed. The *TarBase_microarray* dataset has a limited number of miRNA sequences due to its methodology, which involves measuring expression levels of potential target genes in cells transfected with specific miRNAs. Furthermore, in most datasets (represented by the *h3* dataset curve), a small subset of miRNA sequences (50–60%) are responsible for 90% of the interactions except for the CLIP-based datasets that demonstrate a higher percentage (70–80%).

In addition, we employed KL-divergence as a measure of datasets similarity based on their distributions of miRNA sequences (see Methods: 2.9.3, Fig G in S1 Text). KL-divergence accounts for directionality and quantifies information loss when the source dataset represents the target dataset. We observed a notable divergence between the *TarBase_microarray* dataset and other datasets, both as a target and a source. In addition, a considerable divergence was observed between CLIP-based and CLASH-based datasets. These divergences indicate that datasets poorly represent each other in terms of sequence distributions (Fig G in S1 Text), which can be further confirmed by CDF curves (Fig F in S1 Text), i.e., the number of miRNAs in *TarBase_microarray* dataset is the lowest compared to others. Furthermore, although the CLIP and CLASH datasets originate from the same experiment (e.g., *h3*) in our case, there is a distinction in the nature of the data. The CLASH dataset represents bonafide interactions (chimeras), while the CLIP dataset includes predicted interactions. This discrepancy in the nature of the data contributes to the observed differences in miRNA representation between the two datasets.

**3.2.2 Cross-datasets classification performance.** For each dataset, we loaded the XGBoost classifiers that we trained in subsection 3.1 and used them to classify the test sets of the other datasets (repeated 20 times for each train-test split, Fig 2). The primary measure reported below is mean ACC (Fig 5, Fig H in S1 Text). We provide other performance metrics TPR, FPR, TNR, and FNR in Figs I-L in S1 Text, respectively. The ACC values vary among the

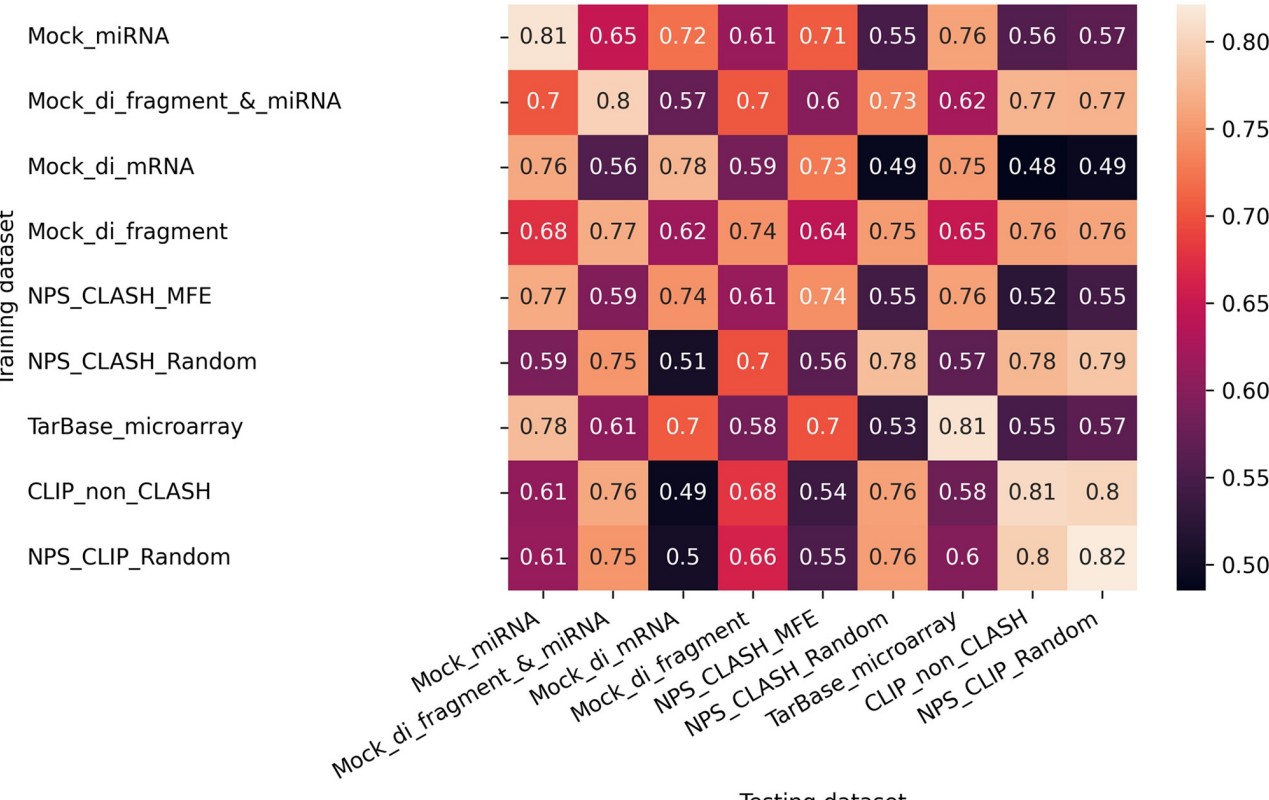

**Fig 5. Cross-datasets classification results.** Cross-datasets classification ACC for pairs of negative datasets. Each cell *(i,j)* represents the mean ACC of the 20 classifiers that were trained on dataset *i* and tested on dataset *j*. The standard deviation (std) values can be found in Fig H in S1 Text. Note that, for ease of interpretation of the results, the color scale is the same as in Fig 3 for intra-dataset analysis and inverse to the scale used for the KL-divergence plot in Fig G in S1 Text. For interpretation of the results, refer to dataset characteristics and feature importance analysis in Figs F, G, and M in S1 Text.

pairs, ranging from 0.48–0.82. The matrix exhibits symmetry, i.e., for most pairs of datasets, the classification performance is similar for *(i,j)* and *(j,i)*, where datasets *i* and *j* serve as training and testing sets, and vice versa, respectively. In general, for most datasets, the intra-classification scores (diagonal) are similar or slightly higher than their scores of cross-classification (either as training or testing dataset) (e.g., *Mock_di_fragment*). However, some datasets, such as *Mock_miRNA* and *NPS_CLIP_Random*, exhibit lower ACC scores in the cross-classification with several datasets than in their intra-classification.

To explain the differences in the performance of the pairs we examined several factors. For example, (*TarBase_microarray, Mock_di_mRNA*) achieved relatively high ACC (0.7) while (*TarBase_microarray, CLIP_Non_CLASH*) achieved low ACC (0.55) scores. First, the KL divergence results indicate that the dataset *TarBase_microarray* is similarly divergent from the other two datasets (Fig G in S1 Text) with scores of 4.4 and 4.2. This observation is further supported by the CDF curves (Fig F in S1 Text), which indicate that miRNA distributions are not the main factor for the variations in ACC results.

Second, in most cases, the distribution of base-pairing patterns of the interactions within the datasets coincides with the ACC results (Fig E in S1 Text). When datasets exhibit a similar proportion of non-canonical interactions with high-density levels, their cross-classification ACC tends to be relatively similar to their corresponding intra-results. The above datasets *TarBase_microarray*, *Mock_di_mRNA*, *CLIP_Non_CLASH* contain 38%, 44%, and 18% non-canonical seeds with high-density pairing, respectively, reflecting that the dataset *TarBase_microarray* is more similar to the dataset *Mock_di_mRNA* in this regard, supporting the higher ACC for this pair compared to the second pair.

Third, features that characterize free energy and miRNA pairing of the interactions (see Methods: 2.5), and are directly related to the density level, were significantly important for both pairs (Fig M in S1 Text). However, the difference lies in their impact on classification success and cross-dataset knowledge transfer. For the first pair, these features contribute to successful classification and are consistent for both datasets. Although energy and miRNA pairing features remain crucial for the second pair, they do not aid in the classification task (low performance), suggesting a disparity in energy distribution between the datasets, which hinders effective information transfer from the training dataset to the testing dataset.

**3.2.3 Two negative-classes classification.** To further understand the variability within cross-dataset results, we conducted additional analysis by mixing two negative datasets *(A,B)*. We then (1) evaluated the ability of a classifier to distinguish between them and (2) identified the features that contribute to the discrimination. The ACC varied among the pairs, ranging from 0.54 to 0.68 (Fig N in S1 Text). Below we focus on two specific pairs *(Tarbase_microarray, Mock_miRNA)* and *(Tarbase_microarray,CLIP_non_CLASH)* that received significantly different ACC in the cross-dataset classification (Fig 5) of 0.78 and 0.55, respectively. Importantly, in both cases, the differences arise mostly from FPR 0.24 and 0.7, respectively, while the FNR is the same (0.196) for both (Figs J and L in S1 Text).

The two-negative classes model of the first pair struggled to effectively distinguish between the datasets (ACC = 0.56, FPR = 0.73, FNR = 0.14), as opposed to the second pair that demonstrated better performance (ACC = 0.65, FPR = 0.46, FNR = 0.23). These performance results reflect the similarity or dissimilarity between the datasets in the first or second pair, respectively, which facilitates or hinders the successful transfer in the respective cross-dataset classification. We generated SHAP plots for both pairs to identify the key features influencing the two-negative class classifiers (Fig O in S1 Text). In the first pair, multiple features showed less clear separation, which explains the lower discriminative ability of the model. Conversely, in the second pair, duplex energy was the dominant feature that significantly impacted the model, accounting for the high ACC of the second pair compared to the first.

### 3.3 One-class classification performance

We investigated whether negative interactions are necessary for the classification of MTIs or if it is feasible to develop models that utilize only one class of available information. To that end, we evaluated the performance of one-class SVM and Isolation Forest algorithms that are trained on positive class examples only. We compared them with their corresponding binary learning models SVM and random Forest, respectively (see Methods: 2.6.2, 2.8), using multiple metrics. As the trends across different negative datasets were comparable (Fig P in S1 Text), we refer in detail only to *TarBase_microarray* dataset (Fig 6). Notably, one-class classifiers demonstrate a nearly perfect TPR. At the same time, the TNR is negligible, implying that these classifiers cannot differentiate between positive and negative interactions and instead categorize all negative interactions as positive.

Isolation Forest achieved a low ACC score of 0.5 while Random Forest reached a high score of 0.78. Additionally, FPR was 0.186 and TPR was 0.814 for the Random forest, implying its proficiency in accurately identifying both positive and negative interactions, which is not reflected in the performance of the Isolation forest. The One-Class SVM and SVM classifiers exhibited comparable ACC scores of 0.49 and 0.57, respectively. However, SVM outperforms one-class SVM in identifying negative interactions, as evidenced by its lower FPR and higher TNR values.

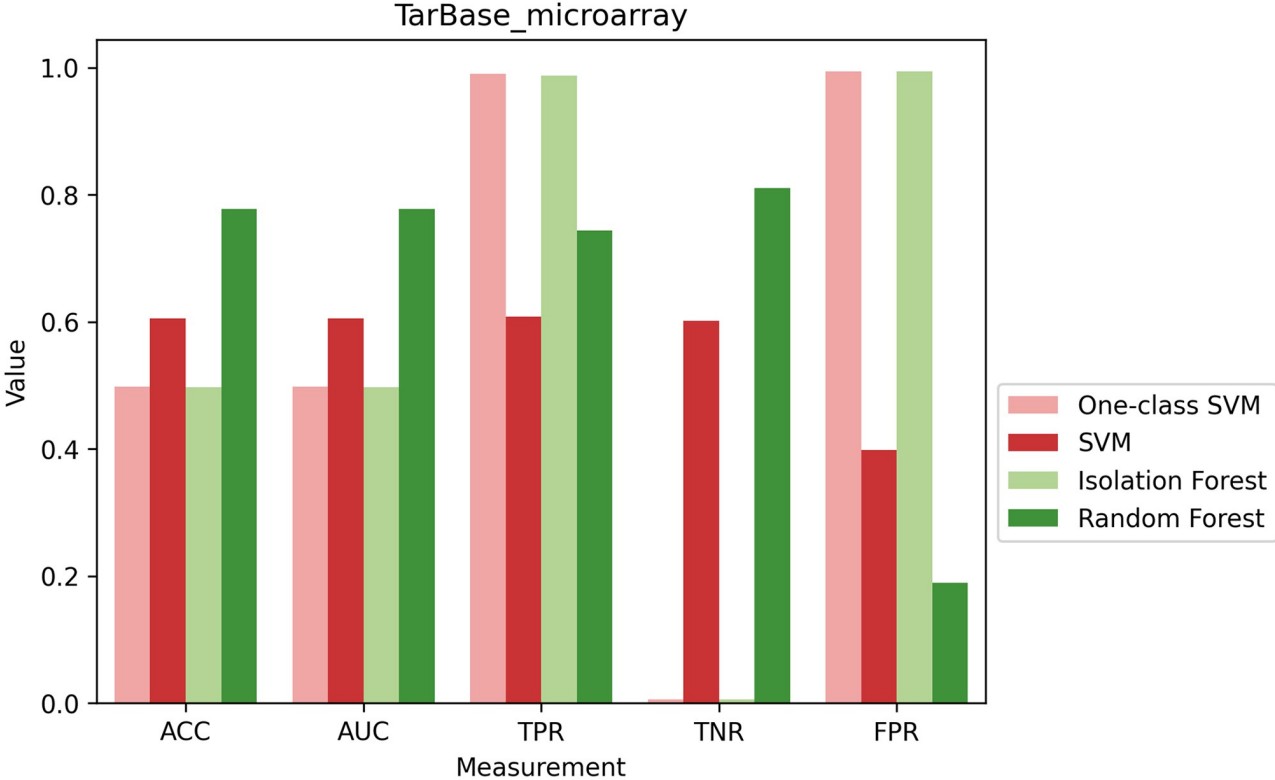

**Fig 6. Comparing the performance of One-class classifiers with their respective binary classifier.** The performance of four classifiers is measured by ACC, AUC, TPR, TNR, and FPR. The One-class SVM and Isolation Forest were trained on positive data only while the SVM and Random Forest were trained on both positive and negative datasets. All four models were tested for their ability to classify both positive and negative interactions. The negative dataset reported here is *TarBase_microarray*. For similar analyses with other negative datasets see Fig P in S1 Text.

## 4 Discussion

Accurately identifying bonafide miRNA targets is essential for understanding their functional roles. Despite advancements in generating high-throughput datasets of MTIs, persisting technical challenges have fueled a growing interest in computational approaches, specifically advanced ML models. In binary classification, the model needs to distinguish between positive MTIs (those recovered in high-throughput experiments) and negative interactions (those lacking binding evidence). The absence of experimentally validated negative MTIs, led to the need to generate them artificially. Each method for generating negative interactions aims to sample interactions from a relevant search space utilizing a specific strategy to identify an initial set of candidate interactions for the negative category.

We hypothesized that selecting the appropriate method for negative datasets is crucial for ML models' classification ability. The ultimate goal of this study was to investigate the impact of different approaches to generating negative data on the classification of true MTIs. To accomplish this objective, we generated datasets of negative interactions using different strategies. We also demanded that interactions meet basic requirements (i.e., canonical or non-canonical base-pairing in the seed region). This filtering sought to achieve a balance and avoid negative examples that are overly distinct or similar to the positive ones, to ensure effective training.

### Intra- and cross-dataset analyses indicate the difficulty of ML models to generalize interaction rules from training to testing sets

Having the negative datasets in hand, we aimed to first assess the quality of the datasets by training/testing the ML model on each one of them and (2) evaluate the generalizability of these models on independent test sets (i.e., other negative datasets). The first task proved difficult, with a maximum ACC of 0.82. This suggests that the negative interactions were of high quality, resembling positive interactions and avoiding excessive artificiality. When examining a specific classifier in its intra- vs cross-classification performance, we observed similar TPRs due to shared positive interactions in the test sets; and these TPRs were relatively high across most classifiers (correspond to rows in Fig I in S1 Text). Nevertheless, significant variations were observed in the FPRs of the same classifier. For example, for *Mock_di_mRNA* FPRs range from 0.19–0.77 (Fig J in S1 Text); highlighting the model's variable risk of misclassification of negative interactions, which is undesirable in real-world applications. Furthermore, almost all classifiers, showed a high FPR on at least one dataset, indicating the inability of the model to generalize interaction rules from learning to testing sets.

### Specific methodological characteristics explain the differences between datasets

The performance of a classifier can be highly affected by methodological decisions that introduce biases into the characteristics of negative datasets. In the case of *mock_mono_mRNA* method, we demonstrated that the nucleotide shuffling method influences the target composition properties. The intra-analysis ACC of this dataset is above 0.9, indicating that the model effectively learned the pattern of a specific small group of dominant features.

Another characteristic that causes differences between datasets is related to the search space of the binding *site* on the target sequence (i.e., experimentally derived *fragment* or on the *full mRNA*). Clustering of the datasets based on their cross-dataset ACC values, clearly shows that ACC values are higher for dataset pairs that use similar portions of the target gene (fragment or full mRNA) to allocate the sites (Fig Q in S1 Text). A larger search space enables

RNAduplex to identify more energetically favorable regions, influencing the energy characteristics of the interactions. Not surprisingly, energy characteristics emerged as the most influential factor in the classification process of negative interactions that were generated using different portions of target genes (Fig R in S1 Text).

In some existing methods, negative candidates are filtered based on specific criteria. In our study, in *NPS_CLASH*, for example, we employed one of two selection options for candidates of specific miRNA-mRNA pair: interaction with the lowest MFE or random selection. The cross-dataset ACC scores of the pair *NPS_CLASH_MFE* and *NPS_CLASH_Random* in both directions are significantly lower than their respective intra-dataset results. These notable disparities arise because energy-based selection favors interactions with biased energy characteristics. To counterbalance this bias, we generated datasets with two additional selection methods that prioritize candidates with relatively good energy properties rather than optimal ones: median MFE and random selection from the top 20th percentile based on MFE. We then applied cross-dataset analysis between all four selection options (Fig S in S1 Text). We found that the selection method impacts the classification performance, as is evident by the lower ACC of the cross-dataset compared to all intra-dataset classifications. Nevertheless, the model trained on *NPS_CLASH_20th_Percentile* exhibited subtle differences in performance, suggesting it better generalized the interaction rules.

The use of artificial or real miRNA/mRNA sequences in negative datasets is another factor to consider. The literature has debated the impact of artificial sequences on the model's ability to differentiate between positive and negative interactions, as it may learn to distinguish between the sequence patterns rather than the interaction patterns [23,28]. Surprisingly, our findings demonstrate that datasets composed of artificial sequences or real sequences achieve comparable ACC in intra-analysis, probably since we do not include sequence-based features [23]. Furthermore, in certain cases, datasets composed of artificial sequences exhibit strong generalization capability on datasets composed of real sequences in the cross-analysis.

## Maintained factors in dataset evaluation

To ensure a rigorous comparison between the methods, we maintained specific parameters throughout our analysis. We used the RNAduplex tool to find the location of the binding site of miRNAs on target sequences. The duplex selection process by this tool introduces bias, as it chooses energetically favorable regions. Consequently, energy-related features influence the classification model. We focused on canonical and non-canonical interactions as these are considered functional. In addition, we employed XGBoost as our ML method. This method was shown to outperform other ML methods in MTI prediction [23] and to achieve better results compared to deep learning models when handling tabular data [52]. Performing similar analyses using different ML and deep learning algorithms, alternative duplex generation tools, seedless interactions, and additional interaction features could be a venue for future research.

Specifically, we demonstrated the analysis utilizing human experimental datasets, with a particular human MTI dataset (*h3*) serving as the positive dataset. Based on our previous research [23, 53], which shows that MTI interaction rules are transferable between evolutionary close species, we speculate that our conclusions– that different methods for negative datasets impact the performance of ML models– could also extend to other species. However, this requires validation in further studies.

## Combination of negative datasets

To further examine the biases in negative data generation, we created a negative dataset composed of a balanced mixture of all the negative datasets proposed in this study and trained/

tested a new model with this data along with the positive set. Interestingly, the model failed to classify the negative interactions correctly. This outcome reiterates the insight that the negative interactions are biased by the specific strategy of each generation method. Consequently, the model that was trained on this combined dataset struggled to identify a consistent pattern that characterized all negative interactions while distinguishing them from positive interactions. It is possible that other strategies, including ensemble methods for combining models trained on different negative datasets, would yield better performance and should be examined in future studies.

## Recommendation for negative data selection and examination of new ML models

The process of generating negative interactions can introduce biases to the data, leading the model to easily distinguish between positive and negative interactions in the training set and then fail on the test set. To avoid such a scenario, we propose some guidelines: (1) Avoid bias in one or more features. For instance, if the k-mer composition is modeled in the features, maintaining the original ratio distribution of k-mer sequence composition in the artificial sequence is crucial, and can be achieved through k-mer shuffling (e.g., Mock mRNA sub-methods); (2) Keep negative interactions as similar as possible to positive interactions (such as maintaining canonical or non-canonical seed) and (3) Generate negative interaction from diverse miRNAs and mRNAs to enhance model generalizability (e.g., *TarBase_Liver* interactions originate from a limited set of miRNAs, and its model failed to generalize and exhibited model overfitting).

To assess the generalizability of a new ML model for MTI classification, it is essential to evaluate its ability to effectively capture positive interactions while minimizing the risk of misclassifying negative interactions as positive. This is characterized by achieving a high True Positive Rate (TPR) and a low False Positive Rate (FPR), respectively. As mentioned earlier, we identified the search space for the binding site on the target sequence (i.e., experimentally derived fragment or the full mRNA) as a key factor contributing to differences between negative datasets. To evaluate the model's generalizability, it is recommended to test it on negative interactions derived from both strategies. Divergent results between these strategies would indicate low generalization, while similar results would affirm the model's effectiveness. Notably, the *Mock_di_fragment* method demonstrated a notable success rate in identifying positive interactions in cross-classification analysis. However, it occasionally misclassified a few negative interactions as positive. Despite this, the method exhibited good generalization overall (Fig Q in S1 Text).

## One-class classification models highlight the need for negative data in MTIs classification task

As an attempt to avoid generating negative datasets for training, we trained One-class SVM and Isolation Forest solely on positive interactions. However, when testing them on both types of classes, we observed very poor classification performances overall. This low performance suggests that these models struggled to learn solely from positive examples, as they misclassified almost all negative interactions, highlighting the challenge of distinguishing between both classes. Consequently, these models, commonly used for anomaly detection tasks to identify outliers, are unsuitable for MTI analysis due to the similarity between positive and negative interactions. This further underscores the importance of negative examples to create accurate classification models that can distinguish between the interactions despite the similarities.

### Extension to other biological interaction data

While our research focuses on miRNA-mRNA interactions, the methodologies and insights gained can be extended to other types of biological interactions, such as Transcription Factor (TF)-DNA and RNA Binding Protein (RBP)-RNA interactions. In these domains, the challenge of generating reliable negative data remains critical for developing accurate predictive models. For TF-DNA interactions, several methods have been proposed for the preparation of negative datasets with low expected TF binding, including sequences not labeled as TF binding sites, downstream random exons, random selection of non-coding sequences, or DNA regions far away from genes [54]. For RBP-RNA interactions, methods for negative data generation include shuffling the coordinates of binding sites within all genes with at least one binding site or extracting sites that are not identified as interacting [55]. Following the results of our study, it is critical to compare and evaluate the impact of these methods on classifier performance and establish guidelines and standards in these fields as well.

## 5 Conclusion

Our research emphasizes the crucial impact of employing diverse techniques for generating negative data on MTI classification tasks. Furthermore, it illuminates how different methodologies might favor interactions with particular characteristics, leading to inherent biases that enable the ML model to discriminate between positive and negative interactions.

In summary, our conclusions underscore the necessity of standardized approaches in future research, fostering comparability and reducing potential biases in MTI classification studies. Adopting such practices will contribute to the advancement of this field, providing a solid foundation for further investigations and the development of more accurate classification models.

## Supporting information

**S1 Text. Supplemental Figs A-S, the description of features, and the pseudocode for the negative dataset generation algorithms.**
(PDF)

**S1 Table. List of features and their definition.**
(XLSX)

## Acknowledgments

We thank Gilad Ben Or for providing the datasets and code generated in [23], and Prof. Lior Rokach and the members of the Veksler-Lublinsky laboratory for helpful discussions.

## Author Contributions

**Conceptualization:** Efrat Cohen-Davidi, Isana Veksler-Lublinsky.

**Formal analysis:** Efrat Cohen-Davidi.

**Funding acquisition:** Isana Veksler-Lublinsky.

**Investigation:** Efrat Cohen-Davidi.

**Methodology:** Efrat Cohen-Davidi, Isana Veksler-Lublinsky.

**Resources:** Isana Veksler-Lublinsky.

**Supervision:** Isana Veksler-Lublinsky.

**Writing – original draft:** Efrat Cohen-Davidi, Isana Veksler-Lublinsky.

**Writing – review & editing:** Efrat Cohen-Davidi, Isana Veksler-Lublinsky.

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
