## [Decision Letter · Decision Letter 0]

3 Dec 2023

Dear Dr Veksler-Lublinsky,

Thank you very much for submitting your manuscript "Benchmarking the negatives: effect of negative data generation on the classification of miRNA-mRNA interactions" for consideration at PLOS Computational Biology.

As with all papers reviewed by the journal, your manuscript was reviewed by members of the editorial board and by several independent reviewers. In light of the reviews (below this email), we would like to invite the resubmission of a significantly-revised version that takes into account the reviewers' comments.

We cannot make any decision about publication until we have seen the revised manuscript and your response to the reviewers' comments. Your revised manuscript is also likely to be sent to reviewers for further evaluation.

Sincerely,

Ilya Ioshikhes

Section Editor

PLOS Computational Biology

Reviewer's Responses to Questions

**Comments to the Authors:**

Reviewer #1: In the manuscript "Benchmarking the negatives: effect of negative data generation on the classification of miRNA-mRNA interactions" Cohen et al address the problem of negative entries, and assess approaches for comparing and contrasting one class classification vs multiple class classification.

The authors focus on animal microRNAs - details presented in the introduction and approaches point in this direction, however, this is not acknowledged by the authors. Moreover the description of the technical, yet biological, characteristics of the miRNA/mRNA interaction are not thoroughly understood and presented; Even though machine learning approaches are highly popular at the moment, I think it is important to thoroughly and correctly present their characteristics, and avoid high level descriptions that are not sufficiently informative to either novices or experts in the field.

The description of the methods is wordy and I find it difficult to extract its essence.

The code is made available on a github repo - this is an excellent initiative; to enhance the visibility and usefulness of the code, I suggest adding mini descriptions and comments to the code.

For the results section, it is important to show standard ML outputs, but simple values such as accuracies are not sufficient to assess a model or interpret its efficiency.

The problem tackled is interesting and relevant.

Unfortunately to approach doesn't capture all the information from the proposed study.

Reviewer #2: General comments

Positive

The interpretation of data was interesting as the authors have made an effort to investigate what the model is learning and not just taking into account accuracy metrics.

Negative

The manuscript is lacking focus at large, following are some potential directions that could make the research presented actionable for the field.

a. The paper would greatly benefit from incorporating an analysis of biases against multiple negative sets.

b. Guidelines for selecting good negatives based on the findings in the paper. Potentially, some guidelines on how to test if a model generalizes well?

c. Authors used only one type of ML model (XGBoost). It would be interesting to see if the selection of ML (or even DL) model has an impact on the results.

d. A method used in the literature for generating negatives where positive miRNA-mRNA pairs are randomly shuffled, maybe with some constrains so that negative pairs are also not in the positive set, should be tested as it could potentially have less bias than other methods presented.

e. The data deposited in Zenodo (https://zenodo.org/records/8199848) are not split into train and test sets making reprodcubility of the method difficult. It would be nice to have it there so results could be reproduced or new models could be trained on exactly the same data.

----

Review comments

Line 154 - 155

Only canonical and non-canonical interactions are considered valid in our study, both in positive and negative datasets.

Why have non-seed interactions not been considered in this study?

Line 200 - 201

We chose h3 as the positive interactions dataset for our experiments.

What is the reasoning to pick h3 as the positive interactions dataset?

Line 205

We collected from the literature methods for generating negative interactions.

The method by which the comprehensive literature search was performed should be explicitly stated in the materials and methods section.

Line 211

(similar to, e.g., [16, 19].

missing closing parenthesis

Line 212

a random shuffling (uShuffle with k=1 [44])

why shuffle only individual nucleotides here, when in method 2 instead both individual and pairs of nucleotides are shuffled?

Line 216

(the number of attempts was limited)

what happens when a valid negative cannot be found after the given number of attempts?

Line 310 - 311

We split the positive and negative datasets into 80% training and 20% testing according to a random state (Figure 2).

Random splitting might not be sufficient for genomic sequences as it might produce data leakage - if we have two pairs of miRNA-mRNA interactions with very similar sequences and one pair goes to positive and one to negative set

Line 311 - 312

To address the class-imbalance problem, we employed the commonly used method of undersampling

what method was used for undersampling - is it just random selection, or something more elaborate?

Reviewer #3: In this paper, authors show the impact of the generation of a dataset of negative miRNA-mRNA interacting pairs on the classification of Mirna-Target Interactions (MTIs), underlying the lack of a reliable benchmark dataset of negative interactions and the difficulty of classification of MTIs using existing techniques proposed for generating negative data. The paper presents a detailed set of experiments, also showing the correlation between several mirna and/or target features used for classifying MTIs and the choice of negative dataset.

The problem is well introduced and explained, and the experimental section is clear and exhaustive. However, I have some concerns mainly related to the description of the datasets used for creating the positive and negative datasets of interacting pairs. The nomenclature used is sometimes misleading, with multiple sets identified by the same term.

Here are my detailed comments:

1) Page 5, line 174. It's not clear to me how the complementary CLIP data differ from dataset h3. Is h3 dataset a dataset of CLASH experiment or CLIP experiments?

2) Page 7, Figure 1.

a. Maybe it's better to add a similar picture to explain how the 3 datasets (CLASH, TarBase, CLIP) are processed.

b. I also suggest naming the positive dataset and the extended positive dataset differently instead of grouping them together (e.g. CLASH is both the name of the whole dataset including h1, h2, h3 and h4 and the name of the positive dataset h3).

c. What does "CLIP U CLASH**" mean? Is it a union of the two sets? And if so, does TarBase/CLASH** correspond to a set difference?

3) Page 7, line 211. “similar to e.g. [16,19]” there is a missing round parenthesis.

4) Page 7, line 213. “at most one match with any known human miRNA”. This looks in contrast with what is reported in lines 210-211 where no matches with seeds of other known mirnas are allowed.

5) Page 7, lines 232-234. One of the 6 datasets is not reported in the final list: "Mock_di_mrna"

6) Page 8, lines 269-270. “Interactions that form… were discarded”. Please motivate this choice.

7) Page 8, lines 273-274. Again, it's not clear whether CLASH refers to all 4 datasets forming the CLASH dataset or just the h3 dataset or the extended positive datasets. I think it's better to choose 3 different names to identify these datasets.

8) Page 9, Section 2.5. The description of features is too short. Please specify better what is a feature and what each category represents. In addition:

a. In Table 4, page 10, "Site accessibility" features are not well explained both in the manuscript and in the supplementary table of features. Can you provide more details about them?

b. Have you considered the possibility to include in your study other features related to the stability of mirna-target duplex, e.g., the melting temperature? Or if you plan to include them in future works.

9) Page 11, lines 321-322. “Thus, we enumerated all the possible pairs of training and testing datasets: (train_i, test_j)”. This sentence is not so clear. I think it means that if we have N negative datasets the pairs are (train dataset 1, test dataset 1), (train dataset 1, test dataset 2),...,(train dataset 1, test dataset N), (train dataset 2, test dataset 1), ..., (train dataset 2, test dataset N) and so on, where training and test sets come from negative dataset.

10) Page 14, Fig. 3. I think it's better to invert the colors in the color scale representing the values of metrics. Usually in a heatmap darker colors are associated to higher values.

11) Page 16, lines 494-495. “while the CLIP and CLASH datasets originate from the same experiment”. I guess here with term CLASH we are talking about the h3 dataset only, which is part of the CLASH dataset. This is related to my comment above.

12) Page 17, Fig. 5. Also for the heatmap matrix of Fig. 5 it would be better to invert the colors in the color scale.

13) Page 20, line 628. “NPS_CLASH_20th_Percentline”. I guess it's "Percentile" and not "Percentline"

**Have the authors made all data and (if applicable) computational code underlying the findings in their manuscript fully available?**

Reviewer #1: Yes

Reviewer #2: Yes

Reviewer #3: Yes

PLOS authors have the option to publish the peer review history of their article (what does this mean?). If published, this will include your full peer review and any attached files.

Reviewer #1: No

Reviewer #2: No

Reviewer #3: No
---

## [Decision Letter · Decision Letter 1]

24 Mar 2024

Dear Dr Veksler-Lublinsky,

Thank you very much for submitting your manuscript "Benchmarking the negatives: effect of negative data generation on the classification of miRNA-mRNA interactions" for consideration at PLOS Computational Biology.

As with all papers reviewed by the journal, your manuscript was reviewed by members of the editorial board and by several independent reviewers. In light of the reviews (below this email), we would like to invite the resubmission of a significantly-revised version that takes into account the reviewers' comments.

We cannot make any decision about publication until we have seen the revised manuscript and your response to the reviewers' comments. Your revised manuscript is also likely to be sent to reviewers for further evaluation.

Sincerely,

Ilya Ioshikhes

Section Editor

PLOS Computational Biology

Ilya Ioshikhes

Section Editor

PLOS Computational Biology

Reviewer's Responses to Questions

**Comments to the Authors: **

Reviewer #1: "MicroRNAs (miRNAs) are small non-codingRNAs that regulate gene expression post-transcriptionally via base-pairing with complementary sequences on messenger RNAs (mRNAs)." The statement is ambiguous. The rules of complementarity and the targeted region of the mRNA differ between plants and animals.

Based on the abstract and introduction it is still not clear whether your approach is presented on either plants or animals. Based on the characteristics of the interactions presented in the results, you are focusing on animal models.

"To train an ML algorithm, there is a need to supply entries from all class labels (i.e., positive and negative)."

Please note that multi-class classification is possible on a range of classic and modern ML approaches.

Moreover, the optimisation of a model can rely on partial annotations i.e. the statement "Currently, no high-throughput assays exist for capturing negative

examples, hindering effective classifier construction." is misleading. You yourselves acknowledge this just few sentences below.

"This binding mode allows one miRNA to potentially regulate several hundreds of target mRNAs and for a single mRNA to be targeted by many different miRNAs."

While it is a challenge to predict the number of miRNA targets, it is agreed within the community that miRNAs do not regulate "several hundreds" targets.

The description of the methods is, in places, sloppy

"First, we removed the adapter sequences (as in [13]) and filtered out reads shorter than 18" - I suppose you mean 18nts?

Also, the 18nts and 25nts cutoffs are arbitrary - the filtering should be data-driven.

"To obtain the mature miRNA name corresponding to the aligned region, we mapped the available mature miRNA sequences from miRBase against the precursors as well and selected the mature miRNA with the highest overlap." there are plenty of frameworks for predicting miRNAs and miRNA hairpins, and avoid being biased by the curated miRbase. Once miRNAs are predicted, it is straightforward to indicate which miRNAs are already annotated in miRbase, which are potentially novel.

The description of the 6 approaches for creating the negative dataset is not clearer.

Try summarising the approach as a mini diagram, or in pseudocode.

The sections 2.9 to 2.12 are unclear; please link those to their usage, and, overall, support the story.

"This implies that 1-mer shuffling does not maintain the original ratio distribution of 2-mer sequence composition in mRNA sequences"

I fail to see how 1-mer shuffling works.

The panels you presented require additional support for the interpretation of outputs.

All results are presented from a purely computational perspective, with no links to the biological information or its interpretation.

From a computing angle, this analysis is yet another example of applying XGBoost on a dataset - the power of your results comes from the link with the features and their subsequent interpretation.

Reviewer #2: The authors replies to my comments have been satisfactory. The changes made have substantially improved the manuscript.

Reviewer #3: Authors have addressed my comments.

**Have the authors made all data and (if applicable) computational code underlying the findings in their manuscript fully available?**

Reviewer #1: None

Reviewer #2: Yes

Reviewer #3: Yes

PLOS authors have the option to publish the peer review history of their article (what does this mean?). If published, this will include your full peer review and any attached files.

Reviewer #1: No

Reviewer #2: No

Reviewer #3: No
---

## [Decision Letter · Decision Letter 2]

27 May 2024

Dear Dr Veksler-Lublinsky,

Thank you very much for submitting your manuscript "Benchmarking the negatives: effect of negative data generation on the classification of miRNA-mRNA interactions" for consideration at PLOS Computational Biology.

As with all papers reviewed by the journal, your manuscript was reviewed by members of the editorial board and by several independent reviewers. In light of the reviews (below this email), we would like to invite the resubmission of a significantly-revised version that takes into account the reviewers' comments.

We cannot make any decision about publication until we have seen the revised manuscript and your response to the reviewers' comments. Your revised manuscript is also likely to be sent to reviewers for further evaluation.

Sincerely,

Ilya Ioshikhes

Section Editor

PLOS Computational Biology

Ilya Ioshikhes

Section Editor

PLOS Computational Biology

Reviewer's Responses to Questions

**Comments to the Authors: **

Reviewer #1: Regrettably, the authors are reluctant to listen to feedback.

Reviewer #4: The authors present a novel position dependent codon usage bias (PDCUB) score which is based on the distribution of the first 100 codons of genes. They then study PDCUB and among others they show that it can be used to predict translation-initiating codons with greater accuracy than other models. They report relation to position specific GC content, signal peptide, and more. They suggest that PDCUB defines a spectrum of translational efficiency profiles of tAI. An inverse correlation was found between PDCUB score and ribosomal occupancy in the early transcript. They also examine the relationship between PDCUB intensity and functional enrichment.

This is a potentially interesting study, but additional analysis is required; in addition, the authors should better review and discuss previous studies in the field to better understand the results and their novelty. 

1)The study is focused on human genome. I expect such a paper to analyze at least a few organisms from different domains of the tree of life.

2)There are many important previous studies that should be cited and discussed as they may explain some of the reported results. For example, GC content can be explained by mRNA folding profiles and the beginning of the ORF is populated by many overlapping signals related to AUGs, mRNA folding, AA distribution and more (PMID: 25505165, PMID: 32151272). 

3)Figure 2. It will be helpful to generate a model (e.g. regressor) based on all models or a partial correlation analysis to understand the independent contribution of the PDCUB when controlling for all other models *together*. 

4)Figure 2C: see comments 2), it is easy to explain this region based on the many signals known to appear there.

5)Figure 2A. It was not clear to me if you also check different lengths of the region or only checked 100 codons ? if not- you should clearly try to optimize the length of the region (check shorter and longer lengths). I believe that this length is organism specific (see comments 1)).

6)“Even though PDCUB only considers the first 300 nt of the coding sequence, we observed improved performance with shorter transcripts.” Is it related to the fact that shorter transcripts are highly expressed ? have less splicing ? 

7)GO term analysis: it will be helpful to understand if the signal of part of it is simply related to the expression levels (e.g. there are terms that tend to includes genes with higher expression and this may be the reason you see the association).

8)“translational valley” – a pattern some-how similar were reported (for example) for mRNA folding (see point 2) and also PMID: 22050731).

9)A general comment: each p-value should be positive. See for example: “PDCUB score for the coding transcriptome and found significant correlation between the two parameters (2 ≈ 0.3, = 0.0, Supplementary Figure 20).”.

10)Ribo-seq (e.g. figure 4 D). I expect to see larger differences among the different quantiles. I suggest to check additional dataset and also try to check the average of a few datasets.

11)“Notably, while Tuller and others have examined the features of translational ramps in terms of complete transcriptomes for a given species, we have investigated individual subgroups of human transcripts and found that there is a spectrum of tAI trajectories”..“Previous studies have been conflicted about the presence of a translational ramp in mammals, with some reporting no ramp [28] and others reporting a distinct ramp [22]. We find that both results are true depending on where one looks within the PDCUB spectrum, with the highest-PDCUB transcripts having both a strong initial increase in and steady-state average tAI (translational highway), intermediate PDCUB transcripts having a weaker ramp or no ramp...” note that previous studies claims that some of the genes (e.g. based on expression and function) have ramp (or stronger ramp) and some don’t and an analysis not only based on all genes has been reported. Please read carefully the papers mentioned in 2), 8) and ref. [22]. You should better describe the novelty of your paper (which of course exist).

**Have the authors made all data and (if applicable) computational code underlying the findings in their manuscript fully available?**

Reviewer #1: None

Reviewer #4: Yes

PLOS authors have the option to publish the peer review history of their article (what does this mean?). If published, this will include your full peer review and any attached files.

Reviewer #1: No

Reviewer #4: No
---

## [Decision Letter · Decision Letter 3]

4 Aug 2024

Dear Dr Veksler-Lublinsky,

We are pleased to inform you that your manuscript 'Benchmarking the negatives: effect of negative data generation on the classification of miRNA-mRNA interactions' has been provisionally accepted for publication in PLOS Computational Biology.

Best regards,

Ilya Ioshikhes

Section Editor

PLOS Computational Biology

Ilya Ioshikhes

Section Editor

PLOS Computational Biology

Reviewer's Responses to Questions

**Comments to the Authors: **

Reviewer #4: No additional comments.

**Have the authors made all data and (if applicable) computational code underlying the findings in their manuscript fully available?**

Reviewer #4: None

PLOS authors have the option to publish the peer review history of their article (what does this mean?). If published, this will include your full peer review and any attached files.

Reviewer #4: No

---

## [Editor Report · Acceptance letter]

21 Aug 2024

PCOMPBIOL-D-23-01224R3 

Benchmarking the negatives: effect of negative data generation on the classification of miRNA-mRNA interactions

Dear Dr Veksler-Lublinsky,

I am pleased to inform you that your manuscript has been formally accepted for publication in PLOS Computational Biology. Your manuscript is now with our production department and you will be notified of the publication date in due course.

With kind regards,

Anita Estes
